# The Moderating Role of Social Capital Between Parenting Stress and Mental Health and Well-Being Among Working Mothers in China

**DOI:** 10.3390/healthcare13020117

**Published:** 2025-01-09

**Authors:** Langjie He, Zheyuan Mai, Xin Guan, Pengpeng Cai, Xuhong Li, Juxiong Feng, Suetlin Hung

**Affiliations:** 1Department of Social Work, Academy of Wellness and Human Development, Faculty of Arts and Social Sciences, Hong Kong Baptist University, Hong Kong, China; 20483198@life.hkbu.edu.hk (L.H.); 24481955@life.hkbu.edu.hk (X.G.); fengjuxiong@life.hkbu.edu.hk (J.F.); 2Department of Social and Behavioural Sciences, City University of Hong Kong, Hong Kong, China; xuhongli-c@my.cityu.edu.hk; 3Trinity Centre for Global Health, School of Psychology, Trinity College Dublin, D02 PN40 Dublin, Ireland; caip@tcd.ie

**Keywords:** working mothers, mental health, social capital, parenting stress, depressive symptoms

## Abstract

Rapid economic growth has transformed social and family structure globally, including in China, over the past few decades. With women’s engagement in the workforce, the dual demand of paid and domestic work has become a major source of stress, particularly among mothers. Working mothers face unique challenges of family obligations blended with contemporary values of women’s career aspirations. Methods: This study recruited 189 participants. This study employed quantitative methods to investigate the relationship between parenting stress and mental health, and the moderating role of social capital, among working mothers in Zhaoqing, Guangdong province of China. Results: The findings suggested a high level of parenting stress among Chinese working mothers. Their stress is associated with depressive symptoms. What should be noted is that social capital acts as a moderator between their stress and depressive symptoms. Discussion: This study reveals the buffering effect of social capital against parenting stress and depression in Chinese working mothers, with an emphasis on the importance of workplace flexibility and social support systems in addressing work–family conflicts. The study enriches the study of the buffer effect of social capital on mental health. This knowledge can inform the development of targeted interventions and support systems to improve working mothers’ overall well-being.

## 1. Introduction

In China, a booming economy and urbanization have resulted in the majority of women gaining employment in the last decade. According to statistics, women are obtaining higher levels of education and entering jobs following economic reforms. 320 million women are employed, accounting for 43.1% of the total working population [1]. Although these modern advancements are taking place, traditional Chinese norms still underline familial responsibilities for women. The conventional gendered division of labour highlights that women should prioritize family care, regardless of their professional commitments. Their success is often judged based on their roles as wives and mothers [2,3]. Nowadays, paid employment is not only for financial earning, but also a pursuit of personal and professional growth. In this context, balancing career development with family duties can be challenging.

Historically, Chinese families have relied on the extended family system, in which grandparents and other relatives help with household chores and looking after children [4]. However, this traditional support system has been weakened due to rapid urbanization and industrialization over the past few decades [5]. The migration of young families to cities for better career opportunities has brought about geographic separation among family members, which weakens traditional family support networks. Previously, extended family members could help to relieve the pressure on parents in terms of work–life balance [6]. When family support is no longer available [7], many working women struggle between taking care of children and advancing their careers. Some employers may prefer to recruit men or unmarried women because they worry about the negative impact on the work performance of female employees due to maternity leave and parenting demand, which can affect their career progression [8].

In the past few decades, the Chinese government has enacted measures to support working mothers, such as maternity leave regulations and childcare options. For example, the national government mandates maternity leave ranging from 98 days to several months [9]. The Chinese government has improved the availability of affordable childcare facilities, including sponsored kindergartens and daycare centres. Nevertheless, the coverage and quality of these childcare services can vary between regions and economic tiers [10].

Given the abovementioned context of working motherhood in China, working mothers still face the dual burden of fulfilling parenting responsibilities and their paid work demands without enough social support. This phenomenon can lead to poor mental well-being and increased parenting challenges. Research indicates that working mothers are at higher risk of stress, fatigue, anxiety, and depression, and are more likely to have depressive symptoms or anxiety [11,12].

Social capital is a concept that has been extensively studied in the social sciences. Bourdieu [13] first defined social capital as “the aggregate of the actual or potential resources linked to possession of a durable network of more or less institutionalized relationships of mutual acquaintance or recognition”. Social capital is recognized as a factor contributing to an individual’s health and well-being. More importantly, increased social support is associated with improved mental health ( [14,15]. Social capital consists of an individual’s social relationships, social support, social networks, and social engagement, fostered through local services, facilities, organizations, or physical resources, that reduce people’s risk of mental disorders, while increasing their resilience, adaptability, recovery, and personal well-being [16,17]. Furthermore, social capital could increase social support and reduce stress and levels of depressive symptoms for working mothers [14,18,19]. In addition, social networks correlate with a sense of belonging and purpose, while social participation is positively associated with building relationships and enhancing problem-solving skills through access to new resources [17]. Studies suggest that social capital positively influences working mothers’ mental health and well-being.

Most of the existing research has delved into the direct correlation between parenting stress and depressive symptoms. However, there has been a lack of in-depth exploration of how social capital can be applied as a moderator to reduce such stress and symptoms. Studying how social capital supports Chinese working mothers in the face of family and work pressures will help to fill this research gap. This knowledge could lay the groundwork for developing better support strategies for working mothers.

## 2. Literature Review

### 2.1. Depressive Symptoms Among Working Mothers

Depressive symptoms are prevalent among working mothers. A prospective study in Malaysia revealed that mothers working after pregnancy are at higher risk for physical and mental health issues. The study indicated a greater prevalence of postnatal depression in the first 6 weeks postpartum [20]. The prevalence of depression was found to be significantly different between working mothers and non-working mothers, with working mothers shown to have a higher risk of depressive symptoms [21].

Research on factors affecting depressive symptoms among working mothers is well developed. Socio-economic factors are proven to affect working mothers’ depressive symptoms. Moreover, the age and number of children are also shown to be important factors in the development of depressive symptoms among those who are working [21,22,23]. Wang [22] suggested that non-standard work schedules increase depression symptoms. Working non-standard hours, defined as working hours outside of the typical 9 a.m. to 5 p.m. (mainly nights, evenings, or weekends), may worsen a working mother’s daily routine and potentially lead to chronic disease and emotional problems. Even worse, it could also reduce her ability to handle job demands and stressors. In particular, less-educated working mothers often have fewer job options and are more likely to accept non-standard hours out of necessity instead of voluntarily. It is important to note that their working schedule exacerbates their vulnerability, as they are less likely to have the flexibility to balance work and family responsibilities effectively [22]. Besides the socio-economic factors, including the age and number of children and working schedule, affecting their depressive symptoms, returning to work after childbirth may be accompanied by changes in workload, the nature of work, or work expectations, which may increase their stress [23]. In addition, this stress can be even more severe if there is no flexibility in their work schedule, such as flexible hours or remote work options [24].

### 2.2. Parenting Stress Among Working Mothers

Deater-Deckard [25] defined parenting stress as “the stress reaction to the demands of being a parent” [26,27]. Research indicates that working mothers experience higher levels of parental stress [28]. Their parental stress comes from the dual burdens of work and caregiving duties. For example, Yeo and Teo [29] suggested that parenting stress emerges from the daily responsibilities and tasks of child-rearing. The individual traits of both parents and children influence the intensity of parenting stress. More importantly, high levels of parenting stress negatively affect parents’ mood, marital satisfaction, and parenting behaviour, and children’s development. In some cases, heightened parenting stress can give rise to psychological conditions such as depressive symptoms and anxiety [30]. Studies have also discovered gender-based differences in parental stress, with mothers report higher levels of parenting stress than fathers [31,32]. Parenting stress is also associated with the availability of childcare resources. It is suggested that parents with limited resources and difficulties caring for their children would seek assistance from community social workers to find the support they need [33]. Another study suggested that strengthening family relationships and expanding the social networks of families in need are important protective factors that can improve parents’ well-being. That is to say, social work services should focus on improving these protective factors, rather than solely addressing the deficits or problems within families [34].

In Chinese societies, characterized by Confucian values, mothers hold key roles in the family, which is a fundamental societal building block. Their unpaid work is vital for maintaining harmony. Going into detail, they help with settling conflicts and handling domestic matters within families. Cultural shifts in favour of gender role flexibility notwithstanding, expectations about motherhood remain. These shifts only mean that mothers are expected to balance both work and family responsibilities simultaneously [35,36].

### 2.3. Parenting Stress and Depressive Symptoms

The literature suggests that heavy burdens are experienced by working mothers in China, who have to combine career development with extensive childcare burdens [36,37]. Chen et al. [37] emphasize the dual risks of work and caring burdens that increase stress and cause mental health problems. This situation is exacerbated by intensive parenting norms, especially among middle-class families, in which there is a demand for high parental involvement alongside pursuing career success [36]. Consequently, such conflicts often create much stress, poorer mental health, and family disharmony. What is more, mothers lacking free time to take care of themselves experience a decline in mental well-being [38,39]. Additionally, chronic stress brought about by the balancing of these responsibilities has been associated with depressive symptoms, which are further worsened by physiological changes such as sleeping problems [40,41]. Empirical studies have suggested that the high prevalence of parental stress in China is significantly associated with parents’ depressive symptoms [42]. Research on Chinese parents experiencing depressive symptoms has found that mothers report higher levels of symptoms than fathers [31,32]. Parental stress and depressive symptoms are influenced by a variety of cultural factors, such as social factors and economic factors. For example, Gao and Li suggested that, according to Confucianism, mothers must balance household and caring duties [43]. In addition, traditional gender roles also have an impact on parenting stress and depressive symptoms. Mothers usually bear the main responsibility for childcare, which puts them at higher risk of stress and depression [44]. It should also be noted that rapid urbanization and migration have further impacted parents, particularly migrant parents, who face economic instability and lack social support. In this context, they may find it hard to access quality childcare services [45]. Wei and Chen [46] suggested that economic pressure leads to parenting stress and depressive symptoms when parents face financial strain, due to the increasing cost of childcare and education.

### 2.4. The Moderation Role of Social Capital

Studies have found that increasing social capital could help to alleviate mental health issues and buffer effects on mental health [47]. Almedom and Glandon [48] analyzed the relationship between social capital and mental health, and concluded that the psychological mechanism of belongingness to neighbourhoods and communities benefits adolescents’ and adults’ psychological and emotional well-being. In the context of depressive symptoms, An et al. [49] further investigated the buffering effect of community social capital on late-life depressive symptoms by examining the stress-buffering model. They claimed that social capital acts as a protective factor against depression in older adults. Other studies examining the impact of social capital on depressive symptoms have had similar findings [50]. Furthermore, in recent studies, scholars have probed into the impact of various aspects of social capital on the mental health of postpartum women [51]. For instance, social participation, community network relationships, and community social capital have been found to be negatively correlated with depressive symptoms during pregnancy and lactation [52,53,54].

Social capital is not just about mutual help; it also helps to reduce loneliness by providing a platform to discuss parental issues [55]. In addition, social engagement allows working mothers to be members of wide social networks that provide them with access to different resources to solve problems regarding both work and family [56,57]. These resources provide feelings of belonging and companionship, while positively influencing psychological health.

Similarly, sharing social capital is of great importance when it comes to balancing work and life. Studies show that through their networks, working mothers gain valuable recommendations for things like childcare options and strategies for work–life balance. These things ultimately help them to meet their dual responsibilities more successfully, reduce their stress, and increase their satisfaction in life [58,59].

According to the buffer theory, it holds that the presence of a social support system help to reduce psychological distress and buffer the effects of stressors [60,61]. This theory positions social support as a crucial moderator that can affect the relationship between stressor and health outcomes [62]. The buffering effect is operated through two mechanisms. Firstly, the social support influences how individuals evaluate potential stressors, reducing their perceived threat level and enhancing their coping capabilities [60]. Secondly, social support can either minimize stress responses or influence physiological processes, thus interrupting the path from stressors to adverse health outcomes [63]. Social support includes perceived support and tangible support received from social networks [64]. Empirical evidence supports these theoretical propositions. For example, the social support buffer could be informal networks (friends and family), which exert a great impact on the working mother’s life, and reduce the amount of work it takes to take care of the child to a great extent [65].

This study aims to quantify the parental stress faced by working mothers, understand the association between parental stress and depressive symptoms, and examine the role of social capital in moderating stress and depressive symptoms.

## 3. Methodology

### 3.1. Participants

The data collection was initiated with the support of the principal of a selected kindergarten in Zhaoqing, China. The per capita disposable income (that is, the average disposable income per resident of a region or country in a given period) of the urban residents in Zhaoqing City is 39,679 RMB, which is similar to the national per capita disposable income of 39,218 RMB [66]. This indicates that its residents’ consumption capabilities and living standards are comparable to, or slightly higher than, the national average. As such, the selection of this city could provide a reference for other medium-sized cities in China.

Data collection was facilitated through collaboration with the selected kindergarten in Zhaoqing. The kindergarten teachers were informed about the questionnaires and the importance of voluntary participation and confidentiality during data collection. Before participation, written informed consent was obtained from all the research participants. The informed consent form indicated the study purpose, procedures, and voluntary nature of participation; furthermore, the participants were informed of the potential risks and benefits and confidentiality measures. Ethical approval was obtained from an ethical review board of a university in China. The research participants were anonymous; the study did not collect any identifiable information from them. Power analysis with G*Power (version 3.1) software indicated that with a sample size of 111, α = 0.050, and a medium effect size f^2^ = 0.300, the achieved power was 0.950. This study utilized a dataset of 331 participants who were parents of children in the kindergarten. We excluded fathers and non-working mothers to specifically focus on our target population; our final analytical sample consisted of 189 working mothers who met all the inclusion criteria. Working mothers with children aged 3 to 6 were asked to complete self-administered questionnaires at the kindergarten, and trained kindergarten teachers answered inquiries raised by parents during the process.

After completing the questionnaires, the teachers collected them in sealed envelopes and handed them over to the research team for analysis. In this study, the social demographic variables consisted of gender, age, income, education, and work schedule. The sample mainly consisted of middle-aged female adults who were married with children. Most of them had received tertiary education. In terms of family income, most of the participants reported middle- to high-income levels. Most of them worked standard schedules, while a smaller portion had non-standard working arrangements. The details of the demographic variables will be explained in finding section.

### 3.2. Measures

*The Parenting Stress Index (PSI-36)* is a streamlined version of the original Parenting Stress Index, a tool extensively used to measure the multidimensional aspects of parenting stress [67]. This condensed version, comprising 36 items, evaluates the complexities and potential strains in the parent–child dyad, focusing on three primary domains: parenting distress, parent–child dysfunctional interaction, and difficult child characteristics [68]. The PSI-36 was utilized to assess the stress of family caregivers of kindergarten children in Guangdong province. It uses a 5-point Likert scale, with higher cumulative scores signifying elevated levels of perceived stress, potentially indicating a strained parent–child relationship or heightened parenting distress. This scale reported an efficient internal consistency with Cronbach’s alpha = 0.949.

*Personal Social Capital Scale (PSCS-16)* is a condensed version of the original 42-item PSCS, which is utilized to assess participants’ bonding and bridging of social capital. This version, which has been validated in the Chinese context [69], employs two 5-point Likert scales to evaluate network size and the perceptions of network members. The total score for the social capital scale is obtained by summing the responses for each item, and ranges from 16 to 80. To put it more specifically, higher scores indicate a greater possession of social capital. This study exhibited strong internal consistency and a high reliability coefficient with Cronbach’s alpha = 0.913 for overall social capital.

*Depression Anxiety Stress Scale* is a reliable and validated tool used to measure symptoms of depression, anxiety, and stress over a week-long period, using a 4-point Likert scale [70]. A Chinese version of this scale has demonstrated substantial internal consistency in previous studies [71]. This study primarily employed the depression sub-scale of the DASS. The total score for the depression sub-scale is calculated by summing the responses to the corresponding 7 items, giving a total score range of 0–21 points. Higher scores are indicative of a significant likelihood of experiencing a higher level of depressive symptoms. A test of the reliability of the depressive symptoms sub-scale revealed good reliability, with Cronbach’s alpha = 0.916.

### 3.3. Statistical Analyses

All the data analyses were conducted utilizing SPSS version 26. In addition to descriptive statistics, a correlation analysis was conducted to explore the correlations between the key variables. A linear regression was performed to identify the factors influencing depressive symptoms. Furthermore, the moderation model was tested to ascertain whether social capital moderates the association between parenting stress and depressive symptoms, utilizing PROCESS model 1. This study adapted the bootstrapping procedure with 5000 bootstrap samples, because this helps to overcome the limitations of smaller sample sizes by creating simulated samples to ensure robust statistical analysis [72].

## 4. Results

### 4.1. Characteristics of Participants

In total, 189 working mothers were included in this study. Among them, 67.7% had a standard work schedule, while 32.3% had a non-standard work schedule. Most of the participants (95.8%) were married. Their age ranged from 24 to 50 years old, with M = 36.5 (SD = 4.70). About three quarters of them had at least two children, and 70.8% had a college education. Table 1 shows the sociodemographic characteristics of the participants.

### 4.2. Correlations Among Depressive Symptoms, Parenting Stress, and Social Capital

The working mothers reported relatively high levels of social capital (M = 49.09, Medium = 48.00, SD = 10.20). Their parental stress scores were relatively high (M = 80.51, Medium = 77.00, SD = 21.65) and their mean score for parental stress was lower than for Korean working mothers of children with disease (Mean = 92.88) (Park and Chae, 2020) [73]. Their depression scores were relatively high (M = 9.73, Medium = 8.00, SD = 4.09), and their mean score for depression was much higher than for females in the global context (Mean = 6.33) (Norton, 2007) [74]. Table 2 displays the correlations among the key variables. There was not a significant link between parenting stress and social capital. Depressive symptoms were positively associated with parenting stress (*p* < 0.01) and negatively associated with social capital (*p* < 0.05).

### 4.3. Factors Associated with Depressive Symptoms Among Working Mothers

This study performed a multiple linear regression analysis. Firstly, all of the control variables (age, educational level, marital status, number of children, family income, and working schedule) were entered as the baseline model. Afterwards, the main predictor variables (parenting stress and social capital) were entered to examine their direct effects while controlling for demographic variables. The multiple linear regression analysis revealed several significant factors associated with depressive symptoms among working mothers (see Table 3). Parenting stress emerged as the strongest positive predictor (β = 0.683, *p* < 0.001, 95% CI [0.110, 0.152]). Social capital demonstrated a significant negative association (β = −0.220, *p* < 0.001, 95% CI [−0.136, −0.046]). Age was also negatively associated with depressive symptoms (β = −0.339, *p* < 0.001, 95% CI [−0.005, −0.002]), marital status (β = −0.123, *p* < 0.05, 95% CI [−2.216, −0.095]), number of children (β = 0.132, *p* < 0.05, 95% CI [0.220, 1.861]), and working schedule (β = 0.129, *p* < 0.05, 95% CI [0.176, 2.125]). Educational level (β = −0.107, *p* = 0.064) and family income (β = 0.007, *p* = 0.906) were not significantly associated with depressive symptoms.

### 4.4. Tests of Moderation

Figure 1 illustrates the moderation effect of social capital on the association between parenting stress and depressive symptoms. The Table 4 displays that the main effects for parenting stress (β = 0.124, SE = 0.010, t = −12.567, *p* < 0.001) and social capital (β = −0.098, SE = 0.022, t = −4.469, *p* < 0.001) were both significant. Furthermore, the interaction between parenting stress and depressive symptoms was significant (β = −0.004, SE = 0.001, t = −3.568, *p* < 0.01), which indicates that the effect of parenting stress on depressive symptoms was moderated by social capital. In addition, we conducted a stratified analysis to further examine the association between parenting stress and depressive symptoms in groups with low vs. high social capital. Working mothers whose scores were above or equal to the mean were considered to have a high level of social capital, whereas those whose scores were below the mean were considered to have a low level of social support. The relationship between parenting stress and depressive symptoms was significant in both the low social capital group and the high social capital group. For working mothers with lower social capital, higher parenting stress was reported to be associated with a greater increase in depressive symptoms (β = 0.681, *p* < 0.001). Although parenting stress was still positively associated with depressive symptoms among working mothers with high social capital, this association was slightly weaker (β = 0.668, *p* < 0.001). These stratified results align with and further support the significant interaction effect found in our moderation analysis (b = −0.004, SE = 0.001, t = −3.568, *p* < 0.01), which indicated that social capital may serve as a buffer against the negative impacts of parenting stress on mental health outcomes.

## 5. Discussion

The findings for the demographic variables and depressive symptoms echo the findings of previous research in Western contexts. Age and number of children are important factors affecting depressive symptoms among working mothers [75,76,77]. The working mothers who worked non-standard schedules reported higher depressive symptoms [22]. Wang [22] suggested that working a non-standard schedule (over nights, evenings, or weekends) may adversely affect mothers’ mental health, which further reduces their capacity to cope with job demands and stressors, particularly for those that are less educated, leaving their families and children vulnerable to various challenges posed by such work schedules. However, since jobs with non-standard schedules offer workplace flexibility, they can also contribute to solving some problems by making it more convenient for workers to care for family members [22]. These jobs are often precarious ones that are characterized by temporary work, involuntary part-time work, economic insecurity, unstable wages, and a lack of workplace protections [62]. Women are always expected to balance career development and family responsibilities, and often find themselves in these precarious roles [78]. These precarious jobs might make working mothers’ burdens worse, which increases their risk of depressive symptoms [79].

Surprisingly, the divorced working mothers had a lower level of depressive symptoms. The dominant social discourse regarding an intact family with heterosexual parents and dependent children is prevalent in China. Nonetheless, the meaning of an intact family changed in the 1990s and 2000s in the Chinese-speaking context, as suggested by research conducted by [80]. It has become evident that the meaning of divorce for women has undergone a substantial change over the past two decades [80]. While traditional beliefs have often stigmatized divorce as immoral and suggested that divorced women cannot find happiness, particularly in the process of deciding whether to seek a divorce, there has been a shift toward a different dominant modern position on divorce [81]. Participants in research have indicated that ending a marriage can be practical when there is no longer a romantic connection between partners, based on the notion that marriage should be a companionate relationship. As such, divorce is nothing to be ashamed of [80]. Beyond cultural shifts of attitudes toward divorce, the role of social capital is critical in buffering to levels of depressive symptoms of divorced working mothers. As mentioned, social capital is the resources and emotional support embedded within social networks. It could be a protective factor that mitigates the psychological impact of divorce (Nakagomi et al., 2020) [82]. Divorced mothers often cultivate their informal social networks, including friends and extended family members (e.g. grandparents). Research has shown that the extended family members support can provide them both emotional and practical support for releasing their psychological burden (Metsä-Simola et al., 2024) [81]. For divorced working mothers, these supportive relationships may act as a buffer against the psychological challenges of managing work-life balance.

Consistent with previous research, working mothers’ parenting stress influences depressive symptoms [38,39]. In traditional Chinese society, motherhood is perceived as a key role, reinforced by Confucian ideals. Confucianism, which is a philosophy that is deeply rooted in Chinese culture, points to the family as the main societal unit for social stability. Therefore, mothers are expected to manage domestic and caring work alongside career development, oftentimes by holding double responsibilities without recognition [43]. Though the China of today has witnessed some elements of modern liberalization of gender roles, the deep-rooted gendered labour division helps in shaping the dominant societal norms and notions of motherhood [35,36].

Working mothers are generally exposed to serious challenges as they attempt to balance their work commitments with their caregiving responsibilities at home [37]. For working mothers who struggle with balancing work versus care in their mothering practises at home, the increased demand of work and care responsibilities may lead to the rising stress levels and other mental health issues. Practical difficulties could include work deadlines, daily childcare, domestic work, and other obligations they must fulfil in private spheres [83]. In particular, with the prevalence of intensive parenting valued by middle-class parents, they must maintain a high level of parental involvement within the family, as well as maintaining the pursuit of career success [36]. Hence, pressure could be placed on working mothers to manage the heavy demands of their careers and family [36,37]. For instance, research suggests that heavy demands on mothers for work and care could increase their stress levels and family conflicts to a great extent [39]. Furthermore, research also suggests that a lack of balance between work and care may limit the time available to mothers for self-care and other leisure time for releasing their stress and handling their mental health issues [38]. To put it briefly, achieving a balance between work and family life is not that easy for working mothers, and may become a major source of stress for them. Thus, a higher level of stress as a result of being a working mother may lead to a higher level of depressive symptoms, which is supported by previous research [30].

More importantly, our findings demonstrate the moderation role of social capital on the interaction between parenting stress and depressive symptoms among working mothers. They highlight that social capital functions as a protective factor, buffering the negative impact of stress on mental health problems. These findings align with previous research suggesting the buffer role of social capital [47,64]. This study not only reveals the buffer role of social capital on the interaction between parenting stress and depressive symptoms, but also sheds light on the situations of the working mothers. By examining the impact of social capital on different aspects of a working mother’s life, this study accentuates the role of connections and networks in supporting working mothers’ mental well-being [53,54]. This study extends the theory of social buffering. Previous research has established the buffering effect of social support in general contexts; this study empirically validates its specific applications in the context of working mothers in an emerging urban city of China (Zhaoqing, in Guangdong province). This emerging urban city is unique, because its traditional social support system is being reshaped by urbanization and economic development [84]. With urbanization, working mothers face distinctive challenges, because they have to navigate between maintaining traditional family roles and adapting to modern work demands [85]. Changes in social support in developing cities make the buffering role of social capital especially significant. Social capital can be understood as having one key outcome: it serves as a significant buffer between parental stress and depressive symptoms among working mothers in a developing city, which confirms the protective role of social capital in maternal mental health.

There are several possible explanations for this. According to [65], informal networks, which include friends and relatives, exert a great impact on the life of a working mother, and dramatically minimize the amount of work that is required to make sure the child is cared for. Social capital is not only beneficial in terms of providing assistance to one another, but it also contributes to a reduction in feelings of isolation by offering a forum in which parents may discuss their concerns [55]. Social involvement, on the other hand, makes it possible for working mothers to integrate into broader social circles. These networks give them access to a variety of resources that can help them to find solutions to difficulties that arise in both their professional and personal lives [56,57]. These kinds of resources, also known as social capital, not only foster a sense of belonging and friendship, but they also positively influence an individual’s psychological well-being. In addition, this study suggests the role of social capital in promoting work–life balance for working mothers [58]. Working mothers gain access to valuable resources through their social networks, such as information about childcare options and advice for managing their professional and personal lives [59]. This support enables them to navigate the challenges of balancing work and family more effectively, which helps to reduce their stress and increase their satisfaction in both domains [59]. In Chinese society, the norms of reciprocity highlight the characteristics of the mutual exchange of resources and information [86]. Such a give-and-take relationship could foster a sense of belonging and mutual help, and, eventually, it could build resilience toward parental stress.

## 6. Conclusions

In conclusion, this study demonstrates the significant moderating role of social capital on the relationship between parenting stress and depressive symptoms among Chinese working mothers. In this sample, parental stress and depressive symptoms were relatively high, but social capital was relatively low. The factors associated with increased depressive symptoms included the age and number of children and working schedule. Being married (versus divorced) was associated with lower levels of depressive symptoms. Our findings revealed that parenting stress was positively associated with depressive symptoms across all participants, and this association was stronger among mothers with low social capital compared to those with high social capital. These results suggest that enhancing social capital is an effective means of managing parenting stress and preventing depressive symptoms.

Our findings underlie the importance of fostering networks and connections to promote maternal mental well-being by identifying and emphasizing the protective role of social capital. However, the translation of the findings into practical strategies required a nuanced understanding of how social capital can be cultivated across diverse context.

Zhaoqing is a city in rapid pace of urbanization, which shared the similarity with the other cities in urbanization. They faced reshape of traditional support systems. It is recommended for local governments and non-governmental organizations to establish community programme encouraging the working mothers to connect with each other. Taking the example of Singapore’s parent support groups [87], it is a platform for parents bridge the gap with other parents, the school and broader community. The parental support groups are institutionalized community structures can foster the social capital by sharing resources, providing emotional support, and collaboration in addressing issues related to their children’s education and well-being [87]. Similarly, working mothers in urban area, local governments can adapt this model by establishing parent support networks in schools, so that the working mothers can participate in peer-sharing sessions and parenting workshops. The government funded NGOs could also host parenting programs to offering spaces for mutual support groups, childcare swaps, and family-friendly events. The parent support networks could help alleviate parental stress by fostering mutual aid and support. These connections serve as a form of social capital to reduce social isolation and creating a sense of community among parents navigating similar challenges. Besides of the mutual support network. The employers play a critical role in initiating the family friendly policies, such as the flexible working schedule and on-site childcare facilities. The local government can legislate and incentivize the flexible working hours, remote work options and maternal leave to enable the working mothers to attend to social networks and community activities. These activities were helpful in helping working mothers navigating the domestic care and work. Furthermore, the local governments provide tax incentives to employers who offer subsidized childcare facilities or partner with local childcare providers to reduce the caring burden of working mothers.

In rural area of China, women engaged with employment in informal settings, working mothers in rural area are tied to their dual roles as domestic carer and income earner [88]. Informal employment is subjected to the lack of institutional support and stability of formal work, which makes social capital an essential resource for these women. Rural areas still hold the traditional social support systems, which they have tightly-knit networks based on kinship or shared traditions in China [89]. The grassroots organizations and local governments should leverage existing kinship and community ties to strengthen informal support networks. For example, the regular community gatherings for parenting mutual support purpose, parenting skills sharing sessions, which are beneficious for resource exchange among rural families. Taking the example of India’s Self-Employed Women’s Association, it is a grassroot organizations organizing women working in informal sectors to collectively address economic, social, and cultural challenges. The key actors included mostly community members, NGOs, and government agencies, they foster networks building for mutual support and providing access to essential resources [90]. In the context of rural area of China, local governments and grassroots organizations can similarly leverage kinship ties and shared traditions to organize support networks for working mothers. Furthermore, governments should provide funding and create legal frameworks to support the establishment and growth of women-led grassroots initiatives. These can empower rural working women to navigate between the informal employment and caring burden while strengthening their social capital.

## 7. Implications and Limitations

The results of this study strengthen professional understanding of the parenting stress of working mothers in Chinese societies, especially by highlighting the influence of social capital and parenting stress on depressive symptoms. This study revealed that working mothers experience the mental health issue of depression, implying that they have a need for more social support and services supporting the caregiving process. Due to the need to balance work demands with family caregiving responsibilities, working mothers encounter a double burden, which can jeopardize their mental well-being [37]. This suggests that more resources and social support are needed [91].

On the one hand, it is advised that flexible work schedules and supportive workplace policies should be put in place for the sake of improving the mental well-being of working mothers [92]. For instance, employer can implement flexible working hours, part-time options, shorten working hours and paid parental leave to support mothers during critical stages of child caregiving. Additionally, the employer could provide on-site childcare facilities or childcare subsidies for alleviating parenting stress. The local government’s role of redistribution of resources is essential and widespread adoption of these policies, particularly the working mothers in low-income or precarious employment. The government can provide financial incentives to encourage employers to implement family-friendly policies, such as tax reduction for the employers who adopted the family friendly policies. Furthermore, the government can directly fund on-site childcare facilities in the workplace and subsidize childcare programs for working mothers across all income levels. On the other hand, social workers should pay attention to improving the accessibility of social services and enhancing the coping strategies of working mothers to reduce their stress [93]. For example, parenting support programs could help working mothers handle parental stress more effectively, such as parenting stress management programs, workshops on mindfulness, stress reduction techniques, and time management.

Besides of the intervention solving the parental stress and depression, the social capital is a powerful buffer, since it provides emotional and practical support that can mitigate depressive symptoms. Policymakers can facilitate social capital accumulation by investing in community centers to conduct parenting workshops, mutual-support groups, and recreational activities for families, for example of Singapore’s experiences in facilitating the working mothers’ peer support. Digital platforms and mobile apps can also be leveraged to connect mothers to local support networks and services, particularly in urban areas where traditional community ties may be weaker. Social worker, as helping professionals, play a critical roles in intervention of social capital building.

Focusing on working mothers, this study probed into the correlation between parenting stress and depressive symptoms, and the role of social capital in moderating their mental well-being. The findings are consistent with the results of previous research, which state that social capital is a factor contributing to better mental health for working women who have parenting duties [94]. Research on mothers’ groups in Western Australia indicated that women could develop social capital by participating in local mothers’ groups. The benefits included improved mental health and social well-being and building a supportive community for mothers [19]. Professional practitioners, such as social workers, can offer guidance and resources to improve mothers’ mental health. They can organize stress management training and support networks, especially in low-income communities [95].

The findings demonstrate that there are significant differences in depressive symptoms among working mothers with diverse backgrounds according to their age and number of children, marital status, and parenting stress. These findings serve as a reminder of the importance of designing social services to help working mothers manage their parenting stress. These services need to take into consideration the well-being of children and the marital status of mothers. In particular, social services should pay attention to developing social capital when providing support, as social services bring about better health outcomes. In this process, social capital helps to improve service utilization and create stronger support networks for mothers [96].

Working mothers’ work and care balance is still outstanding under the influence of traditional family values mixed with neoliberalism. Such a double burden comes from caring responsibilities and increasing demands for survival in a market-oriented economy. It is surprising that divorced working mothers had lower levels of depressive symptoms. One possible explanation lies in the changing societal attitudes and subjective meaning surrounding divorce, which is now seen as more acceptable and practical when a relationship ends [80]. When it comes to policy implications, policymakers must target working mothers in marital relationships who have relatively high level of parenting stress. Future studies should investigate the impact of different family structures, such as single-parent families and post-divorce families performing co-parenting, on parents’ mental health and overall well-being. Such a comparison will help to indicate the typical strains and strengths of a particular family structure.

This study has a few limitations. One of the obvious limitations of the study is sampling. Most of the working mothers in the study had higher incomes (over 5000 RMB per month, 70.4%), and they were more educated, completing tertiary education (70.8%). This limitation may influence generalizability of the findings to populations with different socioeconomic characteristics, such as lower-income working mothers. Working mothers with different socio-economic groups may experience differently on parenting stress, depressive symptoms, and access to social capital. For example, lower-income mothers may face more barriers to accessing supportive networks, which could amplify the relationships between parenting stress and depressive symptoms. In the future, participants would be drawn from diverse income and education levels by stratifying the sample based on socioeconomic status for a representative sample. In addition, a cross-sectional design was used for the study, which limits the ability to draw causal inferences among the variables so as to restricts the interpretation of the moderating role of social capital. Future research should adopt a longitudinal design which involves collecting data from the same participants at multiple time points. It enables researchers to track changes in parenting stress, depressive symptoms, and social capital over time. Lastly, another limitation of the methodology was data collection using questionnaires; these may be subject to social desirability bias, which potentially affects the validity of the responses, since participants might tend to provide answers they deem socially acceptable, rather than their genuine perspectives. Future studies can employ mixed methods to triangulate data and reduce the reliance on potentially biased self-reports. The qualitative interviews will be used to gain richer and more nuanced insights into participants’ experiences, which enable the participants to share their thoughts in a more open-ended and less structured way.

## Figures and Tables

**Figure 1 healthcare-13-00117-f001:**
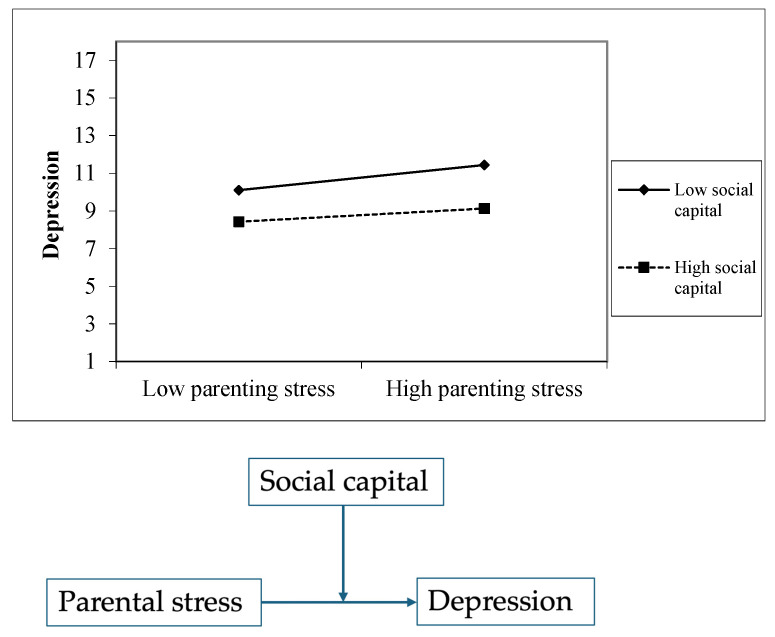
Moderation of the effect of parenting stress on depressive symptoms by social capital.

**Table 1 healthcare-13-00117-t001:** Sample characteristics of working mothers.

Variables	N (Mean)	% (SD)
Age	36.500	4.700
Number of children	1.790	0.520
Marital status		
Married	181	95.800
Divorced	8	4.200
Educational level		
Junior Secondary	6	3.200
Senior Secondary	49	25.900
Tertiary Education (Non-degree/Sub-degree)	67	35.400
Tertiary Education (≥Bachelor’s degree)	67	35.400
Family income (RMB)		
≤3000	6	3.200
3001–5000	22	11.600
5001–10,000	59	31.200
>10,000	74	39.200
Did not answer	28	14.800
Working schedule		
Standard	128	67.700
Non-standard	61	32.300

**Table 2 healthcare-13-00117-t002:** Pearson correlations among key variables.

Variables	1	2	3
1. Parenting stress	1		
2. Social capital	0.043	1	
3. Depressive symptoms	0.652 **	−0.166 *	1

* *p* < 0.05, ** *p* < 0.01.

**Table 3 healthcare-13-00117-t003:** Factors associated with depressive symptoms among working mothers.

	*Depressive Symptoms*		
*Factors*	β	*p*	95% CI
*Parenting stress*	0.683	0.000 ***	0.110–0.152
*Social capital*	−0.220	0.000 ***	−0.136–−0.046
*Age*	−0.339	0.000 ***	−0.005–−0.002
*Educational level* #	−0.107	0.064	−1.096–0.032
Being married (divorced = 0)	−0.123	0.033 *	−2.216–−0.095
*Number of children*	0.132	0.013 *	0.220–1.861
*Family income* @	0.007	0.906	−0.530–0.598
*Working schedule*	0.129	0.021 *	0.176–2.125
*R* ^2^		0.636	
*F for R* ^2^ * change*	36.982 ***		

* *p* < 0.050. *** *p* < 0.001. # ordinal variable. Educational level: 1 = Junior Secondary (reference group). @ ordinal variable. Family income: 0 = ≤3000 (reference group).

**Table 4 healthcare-13-00117-t004:** Association between perceived stress and depressive symptoms, stratified by social capital.

Variable	Low Social Capital (n = 115)		High Social Capital (n = 74)	
	β (95% CI)	*p*	β (95% CI)	*p*
**Perceived stress**	0.135 (0.110–0.161)	<0.001	0.104 (0.072–0.135)	<0.001

Dependent variable: depressive symptoms.

## Data Availability

The data is unavailable due to privacy or ethical restrictions.

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
