# Peer review of "The Moderating Role of Social Capital Between Parenting Stress and Mental Health and Well-Being Among Working Mothers in China"

_healthcare, 2025, doi:10.3390/healthcare13020117_

Round 1

Reviewer 1 Report

Comments and Suggestions for Authors

1. Conclusion and Discussions must be reordered. There should be a section of Discussions, another with Conclusions, another with Implications and limitations section can be included in the conclusions section. 

2. The title "Influencing factors of depressive symptoms" must be improved .

3. Table 3 must have appropriate titles within the table. It is somehow disorganised. *p<.05. **p<.01. ***p<.001. must be under the table. See APA style for table templates. 

4. Mention in Statistical Analysis whether the regression analysis is simple or multiple. 

5. Mention in table 2 that it is s Pearson correlation.

6. In case of regression in results specify how you did it. What type of regression and how you introduced the data in the regression. Were the IV introduced one by one or together?

7. The asteriks in the table 3 must be relocated.

8. At limitations it should be added that the data was collected based on questionnaires and there is a risk of desirable answers. 

9. Include the number of participants at Participants section.

10 . Characteristics of the participants should be included in the Participants section.

11. Why did you use bootstrapping procedure? Explain.

12. Each column in each table should have a title. 

13. To me it seems pretty clear from the theoretical section that there is no surprise of the study. Please indicate why is the study necessary? What brings new? There should be a debate or some contradictory data.

14. You can also include hypothesis not only the purpose of the study. 

15. "To put it more specifically" line 238 delete that, because you add other information that is necessary.

16. The discussion section in abstract "The 19 findings contribute to understanding of parenting stress among Chinese working mothers. The find- 20 ings highlight the role of social capital in reducing depressive symptoms. They underscore the ne- 21 cessity for flexible work policies and increased social support to alleviate the double burden of work 22 and caregiving. This knowledge can inform the development of targeted interventions and support 23 systems to improve working mothers' overall well-being." is too wordy. Be more specific and to the subject. Many ideas repeat. 

Author Response

Thank you very much for your comments, we have addressed each comment and make the revision accordingly. Please find the attached two files for the answers of comment and revised manuscript. Thank you.

Reviewer 2 Report

Comments and Suggestions for Authors

Thank you for the opportunity to review this manuscript. This study explored the moderating role of social capital in the association between parenting stress and mental well-being. The hypothesis is plausible, and the analyses are based on validated measurements. While this is a well-written study, I have some concerns about the analyses.

Please provide the reference number and institution of the IRB approval, as this is a single-anonymous peer review. It should also be clarified whether informed consent was obtained from each of the working mothers.

There is no explanation of how the sample size was calculated. Is this study a secondary data analysis?

In section 3.1, please include the number of mothers recruited, the number who responded to the survey, and the final number of working mothers included in the analysis.

In Table 2, clarify whether the correlations presented are Pearson or Spearman correlations.

In Table 3, educational level and marital status are categorical variables, but the table does not provide the coefficients for dummy variables. Additionally, it is unclear which category is used as the reference category. Please revise this section. For example, divorced status is shown to be inversely associated with depression, which seems counterintuitive. Please review the reference categories for each variable.

For minor formatting consistency, values such as 0.22 should be shown as 0.220, and -0.53 as -0.530.

Replace “P < 0.01” or “P < 0.05” with precise three-decimal p-values.

Despite the interaction effect identified at p < 0.01, Figure 1 shows almost parallel slopes between the two lines. Consider conducting a stratified analysis to examine the association between parenting stress and depression by groups with low vs. high social capital.

Please specify that lines 275-279 refer to Table 3.

In Figure 1, the legend is cut off and reads “Low socia.” This should be corrected.

The statement "The findings indicated a high level of parenting stress among Chinese working mothers" is included, but the results section does not present descriptive statistics for the three main variables. Please include these in the results section, and, if relevant, compare them to the general population of Chinese mothers or working mothers in other countries.

Please consider adding the conclusion section which summarizes the key findings and policy implications

Author Response

Thank you very much for your comment, we have addressed each comment and make the revision accordingly. Please see attached file. Thank you very much. 

Round 2

Reviewer 1 Report

Comments and Suggestions for Authors

1. line 438 "Park & Lee, 2022" and line  410 "Gao and Li, 2021". & or and? Check the whole document. 

2. "Social capitalcapital consists of in" douple word line 70-71

3. line 77-80 "In addition, social networks create a sense of belonging and purpose, while social participation promotes relationships with others and improves problem-solving skills through access to new resources (Trujillo-Alemán et al, 2022). The research studies suggest that social capital can positively impact working mothers' mental health and 80 well-being" be more specific. Use if possible words as "corelates", "influences", "is associated negatively or positively with" which send to statistical measures.

Author Response

  1. line 438 "Park & Lee, 2022" and line  410 "Gao and Li, 2021". & or and? Check the whole document.

Response: Thank you for your suggestion. The format of the in-text citation was revised accordingly.

2. "Social capitalcapital consists of in" douple word line 70-71

Thank you for your suggestion. The double word issues were fixed.

3. line 77-80 "In addition, social networks create a sense of belonging and purpose, while social participation promotes relationships with others and improves problem-solving skills through access to new resources (Trujillo-Alemán et al, 2022). The research studies suggest that social capital can positively impact working mothers' mental health and 80 well-being" be more specific. Use if possible words as "corelates", "influences", "is associated negatively or positively with" which send to statistical measures.

The sentence you mentioned were changed to "Research studies suggest that social capital positively influences working mothers' mental health and well-being." (Please refer to line 80-81). Thank you for your suggestion.

Reviewer 2 Report

Comments and Suggestions for Authors

Thank you for addressing my comments.

Some concerns remain:

1. In Table 4,  beta coefficients are out of the range of 95% CI. Please correct the CIs.

2. The fact that the education and family income was treated as ordinal variables and the reference category for each variable should be described at the table footnote for clarity.

Thank you.

Author Response

  1. In Table 4,  beta coefficients are out of the range of 95% CI. Please correct the CIs.

The beta coefficiency was corrected. The beta coefficiency for low social capital group was .135 (CI: .110 - .161). The beta coefficiency for high social capital group was .104 ( CI: .072 - .135). Thank you for your suggestion.

2. The fact that the education and family income was treated as ordinal variables and the reference category for each variable should be described at the table footnote for clarity.

The "ordinal variable" and reference categories were placed in the table footnote. Please refer to line 341-342. Thank you very much for your suggestion. 

Thank you.